# Thermodynamic Insights of the Molecular Interactions of Dopamine (Neurotransmitter) with Anionic Surfactant in Non-Aqueous Media

**DOI:** 10.3390/ph16091187

**Published:** 2023-08-22

**Authors:** Arshid Nabi, Christopher G. Jesudason, Jamal S. M. Sabir, Majid Rasool Kamli

**Affiliations:** 1Department of Chemistry, University of Malaya, Kuala Lumpur 50603, Malaysia; 2Department of Biological Sciences, Faculty of Science, King Abdulaziz University, Jeddah 21589, Saudi Arabia; jsabir2622@gmail.com; 3Center of Excellence in Bionanoscience Research, King Abdulaziz University, Jeddah 21589, Saudi Arabia

**Keywords:** anionic surfactant, critical micelle concentration, thermodynamic properties, drug–surfactant interactions

## Abstract

This study was aimed at establishing the interactions prevailing in an anionic surfactant, sodium dodecyl sulfate, and dopamine hydrochloride in an alcoholic (ethanol) media by using volumetric, conductometric, and tensiometric techniques. Various methods were utilized to estimate the critical micelle concentration (cmc) values at different temperatures. The entire methods yielded the same cmc values. The corresponding thermodynamic parameters viz. the standard free energy of micellization ($G_{mic}^{o}$), enthalpy of micellization ($H_{mic}^{o}$), and entropy of micellization ($S_{mic}^{o}$) were predicted by applying the pseudo-phase separation model. The experimental density data at different temperatures (298.15 K, 303.15 K, 308.15 K, and 313.15 K) were utilized to estimate the apparent molar volumes ($V_{\phi }^{o}$) at an infinite dilution, apparent molar volumes ($V_{\varphi }^{cmc}$) at the critical micelle concentration, and apparent molar volumes ($\Delta V_{\varphi }^{m}$) upon micellization. Various micellar and interfacial parameters, for example, the surface excess concentration ($\Gamma_{\max}$), standard Gibbs free energy of adsorption at the interface ($\Delta G^{o}{}_{ad}$), and the minimum surface area per molecule ($A_{\min}$), were appraised using the surface tension data. The results were used to interpret the intermolecular interactions prevailing in the mixed systems under the specified experimental conditions.

## 1. Introduction

The main challenge faced by pharmaceutical and industrial scientists is to develop a system that can overcome the low solubility and retain the potency of drugs [1]. In addition, the fine tuning of a drug’s selectivity in vivo is another challenge. In this regard, the techniques of the functionalization and solubilization of drugs with drug–nanoparticles [2] and drugs–micelle delivery systems [3,4] were developed in the past to remove these challenges. The main property of amphiphiles to form self-aggregating micelles with elaborated structures due to their charge polarities and hydrophobic long carbon chains suggests that they could be coupled to drugs to form conjugates [5]. Surfactants or surface-active agents have an affinity to adsorb at the surface or interface, which is accredited to the nature of the solvent and the structure of the surfactant molecule [6,7,8,9,10]. Surfactants have gained prominence in the formulation of many chemical products. Surfactants stabilize emulsions, microemulsions, and nanomaterials and are corrosion inhibitors [11,12,13,14]. The developed drug surfactant conjugates can overcome drug delivery problems and have many pharmaceutical advantages. (i) Micelles improve the bioavailability of poorly soluble drugs. (ii) They can control and maintain the release of drugs within the human body. (iii) The drug–surfactant complex in aqueous and non-aqueous media can be easily prepared and reproduced in large quantities [15,16]. Surface active agents or surfactant molecules have an amphiphilic nature and exhibit the ability of self-organization and self-association as membranes, micelles, or vesicles of fundamental importance in the diverse fields of technical application and biological implementation [17]. The interactions in drug–surfactant systems are essential to understanding the drug action and they provide complete information on more complex biological systems, for instance, the entry of drugs throughout biological membranes [18]. Micelle formation depends on the balance of energetic and entropic contributions due to the molecular interactions of long hydrophobic alkyl chains and the mutual interactions among polar or ionic head groups, solvents, and counterions [19]. In a micellar system, the self-assembly is primarily from the hydrophobic interactions of long carbon chains. On the other hand, van der Waals and hydrogen bonding interactions provide an inevitable contribution to understanding the self-association among the charged molecules in micelles [20,21]. The main aim in pharmaceutical formulation systems is to achieve the drug load and release properties with a long shelf-life and the most minor toxicity. Compared to different drug carrier systems, the surfactant micelles physically entrap the drug molecules inside their core with a minor toxicity, longer residence in the system, and better bioavailability and stability of the drug through micelle assimilation [22,23]. However, besides experimental study, molecular simulation has been used to study the surface interaction of nanoscale black films made from free-standing sodium dodecyl-sulfate (SDS) in terms of disjoining pressure [24]. In addition, the small RGD (1FUV) polypeptide in various environments and the subdomain SD1 of the SARS-CoV-2 spike protein were studied using the ab initio QMRPA method to obtain their respective dielectric spectra, partial charges, and AA bond order [25]. In a recent study, the skin tissue structure was simplified to a graphene surface to eliminate the error caused by skin surface roughness and to more intuitively observe the impact of NaCl concentration and electric field intensity in physiological saline on its infiltration [26].

Considerable efforts have been directed towards drug–surfactant systems in non-aqueous media to obtain a better understanding of the interactions between different species. Our attention is on the sodium dodecyl sulfate (SDS) molecule, an anionic surfactant with an amphiphilic nature [27], which is a proven asset for biotechnological [28], pharmaceutical, and industrial applications. SDS is extensively used in pharmaceutical products, e.g., it is used with ethanol as an antibiotic for skin cleaning, shampoos, and in medicated toothpaste [29]. In addition, SDS molecules are involved in the micellization of triblock co-polymers [30,31]. On the other hand, the dopamine hydrochloride (DA), a neurotransmitter, is chosen because of its essential role in the secretion of hormones inside the central nervous system. The unbalanced secretions of DA result in crucial diseases such as parkinsonism and schizophrenia [32,33]. Ethanol is a polar solvent with self-associating hydrogen bonding. The drug–surfactant system (DA-SDS-ethanol) motivates DA resistance because of its possible implications for solving the problem through the blood–brain barrier [34,35]. Surfactants are amphiphilic in nature, which allows one to observe the affinity of minor molecules to biological membranes. Therefore, the physicochemical interactions of drugs with surfactants can be visualized as an estimation of drug–membrane interactions. Very recently, Zhai et al. reported the possible molecular interactions existing in DA and an ethanol binary system [36]. This is a continuation of our previous studies on the molecular interactions and micelle formation of sodium lauryl sulphate (SLS) with caffeine and theophylline in an aqueous medium at different temperatures (298.15–318.15 K), including the approximate physiological temperature (308.15 K) and micelle formation of SDS with CAD molecules in ethanol media using conductometric and spectroscopic methods [37,38]. Hereby, we augment such studies with a ternary (DA-SDS-ethanol) system. The present study examines the micellization of SDS with ethanol in the presence of DA using surface tension, electric conductance, and volumetric measurements. The study of a surfactant with different drug additives in a solvent environment plays an important role in practical and theoretical grounds. Both the surfactant and the drug have other polarity regions with hydrophilic and hydrophobic characteristics. Such a complex structure is expected to give rise to an investigation of the aggregation behavior in a non-aqueous liquid media, which is one of our objectives.

## 2. Results and Discussion

### 2.1. Conductometric Study

The association behavior of molecules in a diverse fluid system can be imported and inferred using electrical conductivity techniques [6,7]. As a function of SDS in the presence of DA + ethanol, the specific conductivity (κ) was measured in the concentration range from 0.001 to 0.020 mol kg^−1^ at 298.15 K, 303.15 K, 308.15 K, and 313.15 K. This has been summarized in Appendix A and graphically represented in Figure 1.

The cmc values were acquired by the intersection of two line segments above and below the breakpoint (which is the cmc value) of the conductance (κ) against the surfactant concentration at different temperatures. The data presented in Table 1 reveal that the cmc of the SDS in the DA + ethanol mixtures increased with an increase in the temperature from 298.15 to 313.15 K. It has been reported that the effect of temperature on cmc is system dependent [39]. The increased magnitude of the cmc values at higher temperatures reveal the presence of different constituent groups, such as −OH, NH_2_, and the hydrophobic benzene ring of DA, contributing to an interaction with the SDS molecules and causing the destabilization of the micellization with rising temperatures. As is evident with an increase in the temperature, an increase in heat (as a source of energy) was expected with increasing kinetic energies of the SDS surfactant molecules, providing greater chances for interactions of the SDS molecules with the DA molecules, while such collisions caused less aggregations of the surfactant molecules to form micelles. Thus, the expected increased interactions of the SDS with the DA in ethanol and the temperature increase caused micellization destabilization with higher cmc values [40]. The pharmaceutical importance of dopamine and its interactions with ethanol, which reduces alcohol addiction and craving [41], was a motivation for our study of SDS in an ethanol mixture. It was also expected that the higher value of cmc for the SDS with DA + ethanol mixtures may have been attributed to the decreased repulsive electrostatic forces arising from the hydrogen bonding between the −OH groups of the DA and the head group of the SDS molecules. The increase in the cmc values of the drug–surfactant system with an increase in the temperature from 298.15 K to 313.15 K may have been a result of the enhanced solubility of the hydrocarbons and the stabilization of the surfactant monomers [42]; consequently, the hindrances took place in the micelle formation, which resulted in higher cmc values of SDS with DA + ethanol at different temperatures.

It is well documented that the cmc value of an ethanolic solution of SDS at 298.15 K is 8.7 × 10^−3^ mol kg^−1^ [43], which is higher than the observed cmc value of SDS in a DA+ ethanol solution (cmc = 7.90 × 10^−3^ mol kg^−1^) at 298.15 K. The nature of the DA + ethanol mixture plays an important role in the lower cmc value of the SDS as compared to the cmc value of the SDS in pure ethanol at a particular temperature. The possibility of hydrogen bonding between the polar head groups of the anionic SDS molecules and the −OH functional group of the DA molecules is evident, therefore, a decrease in the repulsion forces between the ionic head groups takes place, consequently favoring the micellization of SDS; consequently, the cmc of the SDS in a DA + ethanol mixture decreases. The decrease in the cmc values of the SDS micelles in DA + ethanol may also be because of the solubilization effect of the DA molecules on the SDS between the anionic head groups of the SDS surfactant molecules. This type of solubilization reduces the repulsion between the ionic head groups and favors the micellization of SDS, thus decreasing the cmc values. Shakeel et al. recently revealed in his study that the cmc values of SDS decrease in the presence of levofloxacin + ethanol. However, an increase in these cmc values was observed in an aqueous system, because of the lipophobic desolvation domination over the lipophilic desolvation in ethanol [44]. Hayase and Hayano investigated the effect of 1-alcohols on the cmc of an aqueous solution of SDS and came to the conclusion that the cmc values of the SDS decreased with the addition of 1-butanol, 1-pentanol, 1-hexanol, and 1-heptanol [45]. It is worth mentioning that the cmc values of the SDS in the DA + ethanol (7.90 × 10^−3^ mol kg^−1^, 8.03 × 10^−3^ mol kg^−1^, 8.18 × 10^−3^ mol kg^−1^, and 8.27 × 10^−3^ mol kg^−1^) were less than the cmc values of the SDS in water (8.26 × 10^−3^ mol kg^−1^, 8.40 × 10^−3^ mol kg^−1^, 9.09 × 10^−3^ mol kg^−1^, and 9.90 × 10^−3^ mol kg^−1^) at 298.15 K, 303.15 K, 308.15 K, and 313.15 K [46]. The decrease in the cmc values of the SDS micelles in the present study suggests that the role of relative permittivity has a significant impact on the process of the micellization of the surfactant in different solvent systems.

The thermodynamic parameters associated with micellization, $\Delta G_{mic}^{o}$, $\Delta S_{mic}^{o}$, and $\Delta H_{mic}^{o}$, are also descriptive of the core process in micellization. Changes in these and other related variables are interpreted in terms of changing structural and environmental factors, which directly affects the magnitude of cmc values. In addition, these parameters are utilized to interpret the effects of new environments due to the introduction of other chemical species [47]. Various thermodynamic parameters are deduced by applying the pseudo-phase separation model due to its validity in predicting the energetics of micellization [48]. The standard free energy of micellization is obtained by using the relation:(1)$$\Delta G_{mic}^{o}=(2-\beta )RT\ln X_{cmc}$$
where *X_cmc_* is the cmc values expressed in mole fraction units, *R* is the gas constant (8.314 J K^−1^ mol^−1^), and β is the degree of ionization of the micelles acquired from the ratio of the slopes of two linear segments of the conductivity versus (SDS + DA-ethanol) plots above and below the cmc values [49,50,51]. From the temperature-dependent values of $\Delta G_{mic}^{o}$, we can estimate the other thermodynamic quantities for the instance enthalpy ($\Delta H_{mic}^{o}$) and entropy ($\Delta S_{mic}^{o}$) of the micellization, using conventional thermodynamic relations:(2)$$\Delta H_{mic}^{o}=-RT^{2}(2-\beta ){\left[\frac{d\ln X_{cmc}}{dT}\right]}_{P}$$
(3)$$\Delta S_{mic}^{o}=\frac{\Delta H_{mic}^{o}-\Delta G_{mic}^{o}}{T}$$

The term ${\left[\frac{d\ln X_{cmc}}{dT}\right]}_{P}$ values are determined by fitting $\ln X_{cmc}$ versus temperature with a polynomial function:(4)$$\ln X_{cmc}=a+b(T/K)+c{\left(T/K\right)}^{2}$$
where, *a*, *b*, and *c* are the polynomial constants. Consequently, this arrives at:(5)$$\frac{\ln X_{cmc}}{dT}=b+2c(T/K)$$

It is well reported that, for amphoteric and ionic surfactants, the values of the standard free energy of micellization ($\Delta G_{mic}^{o}$) are in the range from −23 to −42 kJ mol^−1^ at 298.15 K [52]. It is evident from the observed value of the standard free energy of micellization ($\Delta G_{mic}^{o}$), which is −24.97 kJ mol^−1^ for SDS in DA + ethanol at 298.15 K, that this is in this reported range. The standard free energy of micellization ($\Delta G_{mic}^{o}$) values shown in Table 2 are negative at each investigated temperature and, with a rise in temperature, become more negative. This suggests that the micellization of SDS in DA + ethanol mixtures is thermodynamically spontaneous. It is evident from Equation (1) that the values of $\Delta G_{mic}^{o}$ are because of the mutual effect of β and cmc at the given temperature, and the former quantity is found to decrease while the latter one increases with an increase in temperature. As an increase in the cmc disfavors micellization, gradually more negative $\Delta G_{mic}^{o}$ values with a rise in temperature are primarily due to a decrease in the β in a DA + ethanol solution. A decrease in the $\Delta G_{mic}^{o}$ and β, and an increase in the cmc values with the temperature for SDS in lauric acid + DMSO were also reported by Ali et al. [27]. The values of entropy ($\Delta S_{mic}^{o}$) are positive and those of enthalpy ($\Delta H_{mic}^{ο}$) are negative at all the temperatures, suggesting that the micellization is thermodynamically enthalpy- and entropy-driven due to the favorable and changing signs of these terms [53]. The values of the cmc, $\Delta G_{mic}^{o}$, $\Delta H_{mic}^{ο}$, and $\Delta S_{mic}^{o}$ in our study were found to be 0.0080 mol kg^−1^, −27.20 kJ mol^−1^, −2.77 kJ mol^−1^, and 0.0806 J K^−1^ mol^−1^, respectively, at 303.15 K. Earlier, similar behavior was also observed in the case of the drug chloroquine with SDS in a polar aqueous solution, where the cmc, $\Delta G_{mic}^{o}$, $\Delta H_{mic}^{ο}$, and $\Delta S_{mic}^{o}$ values were found to be 0.125 mol kg^−1^, −21.96 kJ mol^−1^, −3.52 kJ mol^−1^, and 60.88 J K^−1^ mol^−1^, respectively, at 303.15 K [54]. The present trend in our data by utilizing the thermodynamic parameters suggests that the hydrophobic and hydrophilic interactions are the major prevailing forces in SDS with DA + ethanol mixtures. The increasing negative values of $\Delta H_{mic}^{ο}$ and the positive values of $\Delta S_{mic}^{o}$ (Table 2) with increasing temperatures effect the onset of the weakening of the hydrophobic interactions between the SDS and DA, while the electrostatic interactions become stronger. The values of $\Delta H_{mic}^{ο}$ are believed to arise from the interactions of the electrostatic, hydrophobic, or polar head group hydrations and counterion micelle bindings [55]. Moreover, the $T\Delta S_{mic}^{ο}$ values are much higher in magnitude as compared to the $\Delta H_{mic}^{ο}$ values, indicating an essentially entropy-driven process, which exhibits the normal mode of micellization [56]. A clear explanation behind this micellization process is the affinity of the structure breaking of the SDS molecules by the ethanol solvent because of the higher degree of rotational freedom of the hydrophobic chains. Thus, this leads to a normal mode of micellization, where the hydrophobic chains of SDS are in the vicinity of the nonpolar interior of the micelles, as compared to the solvent environment [57]. The possible molecular interactions between SDS and DA in the presence of ethanol are hypothetically represented in Figure 2.

### 2.2. Volumetric Study

The experimental density, *ρ*, values of the SDS with DA + ethanol were measured in the concentration range from 0.001 to 0.012 mol kg^−1^ at 298.15 K, 303.15 K, 308.15 K, and 313.15 K and are summarized in Appendix A. The density-dependent parameters were carried out for the system of SDS with DA + ethanol at various temperatures of 298.15 K, 303.15 K, 308.15 K, and 313.15 K and are presented in Table 2. The close perusal of the component structures of this ternary system (DA-SDS- ethanol) reveals that the volumetric behavior of the SDS in DA–ethanol could from different possible interactions. The possible interactions are from (i) the ion–hydrophilic interactions between the charge centers of the −O (SO_4_) head groups of the SDS and the −OH of the para-OH of the DA groups. (ii) The hydrophobic–hydrophilic group interactions between the hydrophobic tail groups of the SDS and the hydrophilic groups of the DA. (iii) The hydrophobic–hydrophobic group interactions between the long-carbon-chain hydrophobic groups of the SDS and the hydrophobic benzene ring of the DA. The temperature-dependent molecular interactions among the components of the mixed systems have been estimated using certain known thermodynamic parameters of micellization, such as in Equation (9). The evaluation of these parameters helps us to determine which molecular interactions are prevailing in any mixed system. In particular, the cmc values may also be determined volumetrically and from the surface tension measurements, complementing and verifying the conductivity approach above. In this regard, the density-derived cmcs of the SDS in DA + ethanol mixtures at various temperatures are shown in Figure 3.

The apparent molar volumes ($V_{\phi }$) were calculated for the SDS with DA + ethanol mixtures using the following equation:(6)$$V_{\phi }=\frac{M}{\rho }-\frac{10^{3}(\rho -\rho^{o})}{\rho \rho^{o}m}$$
where *M*, ρ, and $\rho^{o}$ represent the solute (SDS) molar mass, the density of the solution, and the solvent (DA + ethanol) density, respectively.

The apparent molar volume values, $V_{\phi }$, are positive and increase with the temperature; they are summarized in Table 2. These values are in accordance with those of other systems [58]. In the pre-micellar region, the increasing molar volume could be attributed to the large mean free distance between the molecular outers and an average energy potential. In a micellar region, the increasing temperature would commence a decrease in the micellar concentrations. Thus, these decreased micellar concentrations are due to the dispersion commenced by the strong electrostatic SDS–DA interactions, leading to an increase in the $V_{\phi }$ values. The surfactant in the pre-micellar region is a monomeric 1:1 electrolyte [59]. Thus, the $V_{\varphi }^{{}^{\circ}}$ values were calculated using the Debye–Huckel limiting law as:(7)Vϕ=Vϕo+Avm12+Bvm+……
where $V_{\varphi }^{{}^{\circ}}$, $A_{v}$, and $B_{v}$ are the apparent molar volumes at an infinite dilution, the Debye–Huckel limiting law coefficient, and an adjustable parameter that measures the deviations from the limiting law. In the micellar region, the $V_{\varphi }^{{}^{\circ}}$ values are fitted to the equation as [60]:(8)$$V_{\phi }=V_{\phi }^{cmc}+\frac{\Delta V_{\phi }^{m}(m-cmc)}{\left[B+(m-cmc)\right]}$$
where *B* is an adjustable parameter without a physical meaning, and the apparent molar volume at cmc is $V_{\phi }^{cmc}$. The $\Delta V_{\phi }^{m}$ values are calculated from the difference in the limiting values of $V_{\phi }$ and $V_{\varphi }^{{}^{\circ}}$, as per the equation:(9)$$\Delta V_{\phi }^{m}=V_{\phi }^{cmc}-V_{\phi }^{o}$$

The $V_{\varphi }^{{}^{\circ}}$ and $V_{\phi }^{cmc}$ values are presented in Table 2. All the values increase with an increase in the temperature. This behavior is frequently observed and is attributed to the increase in the thermal energy. For instance, due to the emission of some SDS molecules from the micelles to the bulk solution with increasing temperatures, we observe an increase in the values for parameters ($V_{\varphi }^{{}^{\circ}}$ and $V_{\phi }^{cmc}$) in Table 2. The observed decreasing values of $\Delta V_{\phi }^{m}$ are due to the micelle formation and better interactions between the component molecules in the present study. Similar behavior has also been found in case of the $\Delta V_{\phi }^{m}$ values, reported by De Lisi et al. [61] for the ternary systems of sodium octyl, decyl, and dodecyl sulfates against temperatures at different pressures, attributing the negative slope to the expansibility of the surfactant in the micellar state being smaller than that in the aqueous phase. Ali et al. observed a similar trend in the $\Delta V_{\phi }^{m}$ values for a lipid surfactant system in a non-aqueous media [62]. The observed behavior of these parameters has an essential role in interpreting the micellization process in the studied system.

### 2.3. Surface Tensiometric Study

The surface tension (γ) measurements for the concentration range between 0.001 and 0.020 mol kg^−1^ were carried out at 298.15, 303.15, 308.15, 313.15, and 318.15 K and are summarized in Appendix A. The variations in the surface tension (γ) with respect to the logarithm of the SDS in a 0.05 mol kg^−1^ DA + ethanol mixture at different temperatures are represented in Figure 4.

It is clear that the surfactants reduced the surface tension (γ) of the studied solvent system. The observed results clearly demonstrate that the surface tension (γ) shapely decreased with an increase in the SDS concentration and, apparently, above a certain surfactant concentration, the surface tension (γ) values were nearly constant. This abrupt change in the surface tension (γ) values above a certain SDS concentration is called the critical micelle concentration (cmc), and above this molality, the measured surface tension of the studied system was approximately constant. The adsorption of SDS at the air/solution interface saturated, as the micelle formation took place on increasing the surfactant concentration, which existed at the same time as monomers of the SDS surfactant. The surface excess concentration ($\Gamma_{\max}$), standard Gibbs free energy of the adsorption at interface ($\Delta G^{o}{}_{ad}$), and the minimum surface area per molecule ($A_{\min}$) for the mixed system of SDS with DA + ethanol were calculated. The SDS molecules adsorbed at the air–solution interface with an overall decrease in the surface tension of the solution. The maximum adsorption at cmc at the air–solution interface of the surfactant was measured using the surface excess concentration ($\Gamma_{\max}$) by utilizing the Gibbs adsorption isotherm equation [63]:(10)$$\Gamma_{\max}=\frac{1}{2nRT}{\left(\frac{\partial \gamma }{\partial \log C}\right)}_{T,P}$$

The minimum area per head group of molecules $A_{\min}$ can be estimated by:(11)$$A_{\min}=\frac{10^{18}}{N_{A}\Gamma_{\max}}$$
where *R* is a universal gas constant (*R* = 8.314 J mol^−1^ K^−1^), *N_A_* is Avogadro’s number (6.022 × 10^23^), and *n* is the number of ionic species of the surfactant at the interface. The value of *n* differs with the surfactant bulk concentration and is taken as two, as per previous reports [64]. The $\Gamma_{\max}$ values present the surfactant adsorption tendency or adsorption efficiency at the molecular interface with units in mol m^−2^ (moles per square meter). The units of $A_{\min}$ are expressed in nm^2^ (square nanometer per molecule), while γ is the surface tension and *C* expresses the surfactant concentration.

The excess surface pressure values were obtained at the cmc ($\Pi_{cmc}$) by using the equation:(12)$$\Pi_{cmc}=\gamma_{o}-\gamma_{cmc}$$
where $\gamma_{o}$ is the surface tension of the solvent and $\gamma_{cmc}$ is the surface tension of the surfactant solution at cmc. The Gibbs free energy of the micellization was obtained from the surface tension data and mole fraction units ($X_{cmc}$) using an expression:(13)$$\Delta G_{m}^{o}=RT\ln X_{cmc}$$

Finally, the Gibbs free energy of the adsorption ($\Delta G_{ad}^{o}$) was determined from:(14)$$\Delta G_{ad}^{o}=\Delta G_{m}^{o}-\frac{\Pi_{cmc}}{\Gamma_{\max}}$$

A small discrepancy in the cmc values was noted from the surface tension and volumetric study as compared to the conductmetric determinations. This may have been due to the fact that the cmc is technique dependent. The process of micelle formation is not rapid and it is a well defined process, that is, the cmc spans a narrow range of concentrations because the different steps of micellization are monitored by different techniques. The process of micellization, in general, depends on two major aspects. (i) The hydrophobic–hydrophobic interaction of the long hydrocarbon tail groups of the surfactants. (ii) The electrostatic or hydrogen bonding between the charged head groups of different molecules. Table 3 presents the values acquired by the surface tension data at different temperatures. The values obtained in the present studied systems were influenced by factors (i) and (ii) [65]. The increase in the negative values of $\Delta G_{ad}^{o}$ with a rising temperature indicates that the adsorption of the surfactant molecules at the air/mixture interface took place simultaneously. It also triggered the dehydration of the hydrophilic groups, which are essential in favor of the adsorption process. The surfactant molecules were not so hydrated, so little energy was needed for the process of adsorption to take place with a rising temperature. The observed negative values of $\Delta G_{ad}^{o}$ were much greater than the corresponding values of $\Delta G_{m}^{o}$, which considerably explains the micelle growth, and additional efforts are required for transferring monomeric surfactant molecules at the surface towards a micellar state in a mixed solvent media [66]. Furthermore, the molar Gibbs energy at cmc ($\Delta G_{\min}$) at the maximum adsorption attained is calculated using an equation:(15)$$\Delta G_{\min}=\gamma_{cmc}A_{\min}N_{A}$$
where $\gamma_{cmc}$ is the value of the surface tension at the cmc. The $\Delta G_{\min}$ values are associated with the stability surface with fully adsorbed amphiphilic molecules. The smaller the values of $\Delta G_{\min}$, the more thermodynamically stable surface is formed. In the present study, the surface formed for the SDS with DA + ethanol was more thermodynamically stable at a lower temperature.

## 3. Materials and Methods 

### 3.1. Materials

Analytical-grade dopamine hydrochloride and sodium dodecyl sulfate were purchased from Merck and their purities were specified by the supplier. No additional purification was carried out. These chemicals were dried in a vacuum oven; furthermore, they were kept in a vacuum desiccators over P_2_O_5_ (H_2_O absorbent) at room temperature for about 72 h before use. The ethanol used in our study was of an analytical reagent grade and acquired from Merck. The water content of the ethanol was provided by the supplier (determined by Karl Fisher (K.F.) titration), and its purity analysis was carried out using Gas chromatography (GCMS-QP 2010 Plus, Shimadzu, Kyoto, Japan). Detailed information on the chemicals utilized throughout the work is summarized in Table 4.

### 3.2. Methods

Stock solutions of DA 0.05 mol kg^−1^ in ethanol were prepared by stirring the mixture solution for about 8 h at room temperature. From the stock solution (DA–ethanol solvent), mixture samples of 0.001, 0.002, 0.003, 0.004, 0.005, 0.006, 0.007, 0.008, 0.009, 0.010, 0.011, and 0.012 mol kg^−1^ SDS solutions were prepared for the experimental investigations.

The electronic balance (Shimadzu AY220, Chiyoda-ku, Tokyo, Japan, ±0.0001 g precision) was used to weigh the freshly prepared samples. Air-tight glass bottles were used to avoid evaporation, inducing concentration accuracies. The relevant molecules are systematically represented by their chemical structures in the figures (Figure 5a. sodium dodecyl sulfate, Figure 5b. dopamine hydrochloride and Figure 5c. ethanol).

The prepared samples’ conductance was measured using a digital conductivity meter (PC 510 Bench/Conductivity Meter, EUTECH instruments) with a cell constant of 1.0 cm^−1^. In the present conductivity experiments, the instrument worked on alternating currents (AC) (110–120 V feeding voltage and 50–60 Hz frequency) with an operational frequency of 1 kHz. The instrument was calibrated using standard (Potassium Chloride, KCl) solutions with a conductivity of 1413 µS/cm at 298.15 K. A thermostatic water bath controlled the temperature. The uncertainty associated with the conductivity measurements was estimated as ±0.5%.

The solution densities were measured with an analytical digital density meter DDM 2910 (Rudolph Research, Hackettstown, NJ, USA) at temperatures of 298.15, 303.15, 308.15, and 313.15 K. The proper calibration of the instrument was achieved with doubly distilled water and the density uncertainty was found to be ±0.0001 G cm^−3^. A KSV sigma 702 tensiometer with a Pt-Fe ring was used for the surface tension measurements at different temperatures (298.15–313.15 K). The Pt-Fe surface tension measurement ring was cleaned with a propane flame and washed with de-ionized water. The surface tension measurements of pure water and methanol were carried out at 298.15 K to maintain the accuracy of the surface tension measurements, and they were found to be 71.9 and 22.3 mN m^−1^, respectively. At least three successive measurements were carried out for each sample until the values were reproducible. The uncertainty in the calculated surface tensions was estimated to be ±0.1 mN m^−1^. 

## 4. Conclusions

In our present research work, the thermodynamics of the micellization of a drug–surfactant system were estimated using surface tension, conductivity, and volumetric measurements over temperature range of 298.15–313.15 K. The temperature effect on the cmc values was investigated and different thermodynamic parameters were estimated to examine the structural effect of the surfactant and the studied drug on the micellization. The calculated cmc values of the SDS in DA + ethanol mixtures were nearly similar when measured using different techniques. Various thermodynamic parameters, such as the apparent molar volumes ($V_{\phi }^{cmc}$) at the cmc, apparent molar volume ($V_{\varphi }^{{}^{\circ}}$) at an infinite dilution, and apparent molar volumes ($\Delta V_{\phi }^{m}$) upon micellization, $\ln X_{cmc}$, β, $\Delta G_{mic}^{o}$, $\Delta H_{mic}^{ο}$, $\Delta S_{mic}^{o}$, $T\Delta S_{mic}^{ο}$ and $\Gamma_{\max}$, $A_{\min}$, $\Pi_{cmc}$, and $\Delta G_{\min}$, of the micellization were estimated at different temperatures. The increased positive values of $\Delta H_{mic}^{ο}$ and decreased negative values of $\Delta S_{mic}^{o}$ for the conductivity measurements indicate that the hydrophobic interaction between the SDS and DA molecules weakened while the electrostatic interaction became stronger with rising temperatures. The increased electrostatic and hydrogen bonding interactions with elevated temperatures led to a destabilization of the micellization with higher values of the cmc. The observed positive values of the entropy ($\Delta S_{mic}^{o}$) and negative values of the enthalpy ($\Delta H_{mic}^{o}$) of the micellization at individual temperatures indicate that the process was thermodynamically enthalpy- and entropy-driven. In addition, the observed decreasing values of $\Delta V_{\phi }^{m}$ were due to the micelle formation and better interactions between the component molecules in the present study. In addition, the surface formation for SDS with DA + ethanol in the current study was thermodynamically more stable at lower temperatures, as deduced from the minimum observed value of $\Delta G_{\min}$. The overall results demonstrate that the hydrogen bonding between the −OH group of the DA molecules and the negative head groups of the SDS surfactant molecules in a non-aqueous solvent was dominant. The hydrogen bonding interactions increased with a rising temperature and caused the destabilization of the micellization, with high values of cmc in the present studied system.

## Figures and Tables

**Figure 1 pharmaceuticals-16-01187-f001:**
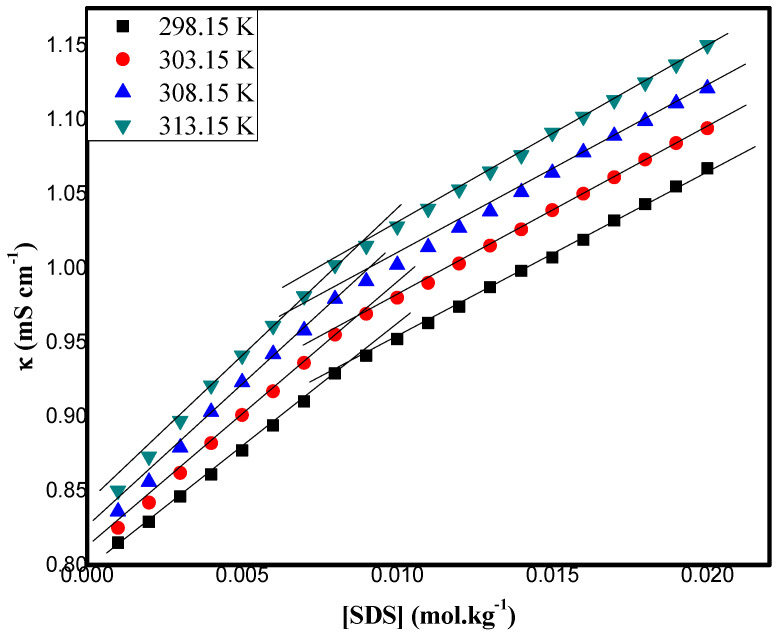
Plots of conductivity, κ, vs. concentration of SDS in 0.05 mol kg^−1^ DA + ethanol at different temperatures.

**Figure 2 pharmaceuticals-16-01187-f002:**
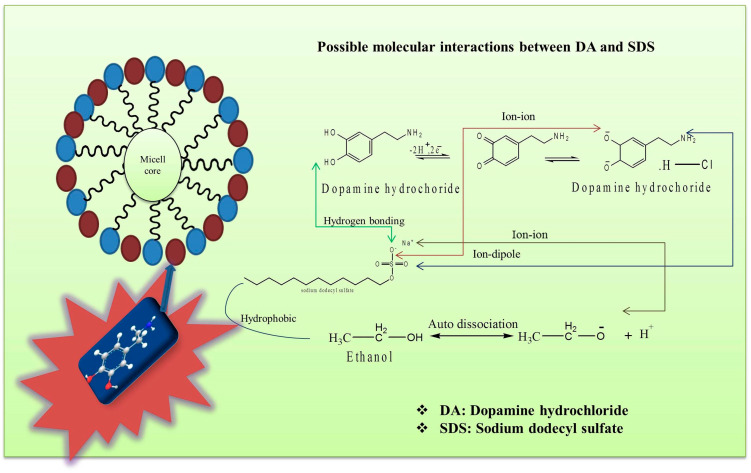
Schematic illustration of intermolecular interactions between SDS with DA in presence of ethanol solvent.

**Figure 3 pharmaceuticals-16-01187-f003:**
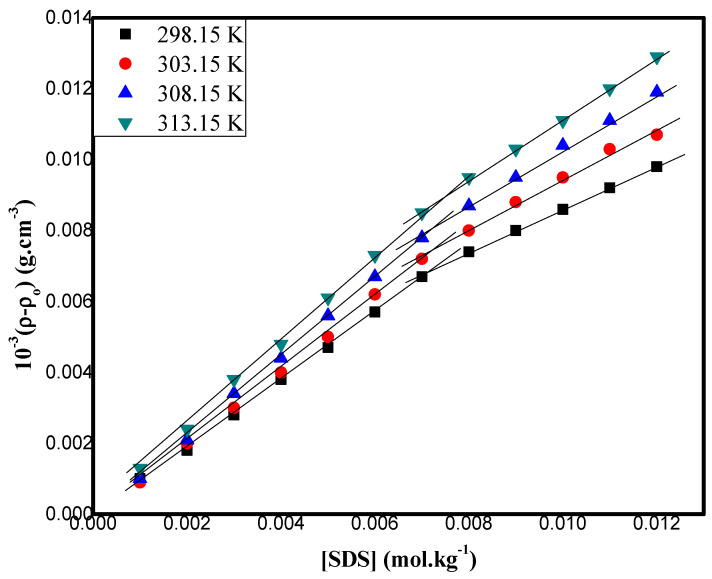
Plots of density, $\rho -\rho^{o}$, vs. concentration of SDS in 0.05 mol kg^−1^ DA + ethanol at different temperatures.

**Figure 4 pharmaceuticals-16-01187-f004:**
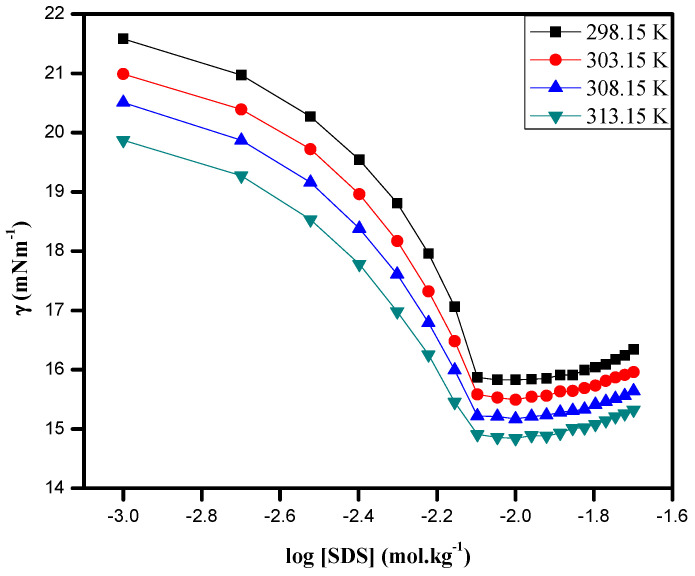
Plots of surface tension, γ, vs. log [concentration] of SDS in 0.05 mol kg^−1^ DA + ethanol at different temperatures.

**Figure 5 pharmaceuticals-16-01187-f005:**
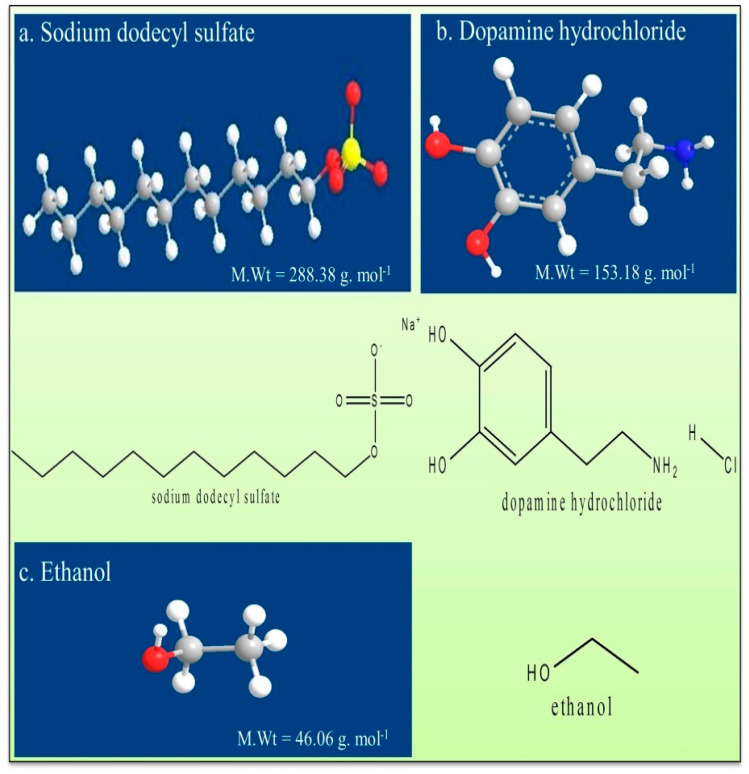
Systematic illustration of chemicals with molecular weights (**a**) sodium dodecyl sulfate, (**b**) dopamine hydrochloride, and (**c**) ethanol.

**Table 1 pharmaceuticals-16-01187-t001:** Values of cmc (critical micelle concentration), *β* (degree of ionization of micelles), ln*X_cmc_* (cmc values expressed in mole fraction units), $\Delta G_{mic}^{o}$ (standard Gibbs free energy of micellization), $\Delta H_{mic}^{ο}$ (enthalpy of micellization), and $\Delta S_{mic}^{o}$ (entropy of micellization) of SDS in 0.05 mol kg^−1^ DA + ethanol at *T* = (298.15 to 313.15) K and 0.1 M Pa pressure.

Parameters	298.15 K	303.15 K	308.15 K	313.15 K
cmc/(mol kg^−1^)	0.00790	0.00803	0.00818	0.00827
*β*	0.7171	0.6225	0.5797	0.5594
ln*X_cmc_*n *X*_cmc_	−7.8525	−7.8362	−7.8177	−7.8067
Thermodynamic parameter of micellization				
$\Delta G_{mic}^{o}$ (kJ mol^−1^)	−24.971	−27.207	−28.447	−29.280
$\Delta H_{mic}^{ο}$ (kJ mol^−1^)	−1.610	−2.777	−4.012	−5.307
$\Delta S_{mic}^{o}$ (J mol^−1^ K^−1^)	0.0784	0.0806	0.0793	0.0766
$T\Delta S_{mic}^{ο}$ (J mol^−1^ K^−1^)	23.360	24.430	24.434	23.973

Standard uncertainties, u, are u(*T*) = 0.01 K, u(*p*) = 0.002 MPa, u(cmc) = 0.0002 mol kg^−1^, u($\Delta G_{mic}^{o}$) = 0.03 kJ mol^−1^, u($\Delta H_{mic}^{ο}$) = 0.03 kJ mol^−1^, u($\Delta S_{mic}^{o}$) = 0.02 J mol^−1^ K^−1^, and u($T\Delta S_{mic}^{ο}$) = 0.04 J mol^−1^ K^−1^ (level of confidence = 0.68).

**Table 2 pharmaceuticals-16-01187-t002:** Critical micelle concentration (cmc), apparent molar volumes at infinite dilution ($V_{\phi }^{o}$), apparent molar volumes at the cmc ($V_{\varphi }^{cmc}$), and apparent molar volumes upon micellization ($\Delta V_{\varphi }^{m}$) of SDS in 0.05 mol kg^−1^ DA + ethanol at *T* = (298.15 to 313.15) K and 0.1 M Pa pressure.

Volumetric Parameters	298.15 K	303.15 K	308.15 K	313.15 K
cmc/(mol kg^−1^)	0.00758	0.00766	0.00779	0.00788
$V_{\varphi }^{{}^{\circ}}$/10^−3^ (m^3^ mol^−1^)	0.2020	1.7838	3.8726	6.0317
$V_{\varphi }^{cmc}$/10^−3^ (m^3^ mol^−1^)	−0.8284	−0.2058	0.3022	0.8184
$\Delta V_{\varphi }^{m}$/10^−3^ (m^3^ mol^−1^)	−1.0304	−1.9896	−3.5704	−5.2134

Standard uncertainties, u, are u(*T*) = 0.01 K, u(*p*) = 0.002 MPa, u(cmc) = 0.2 × 10^−3^ mol kg^−1^, u($V_{\varphi }^{{}^{\circ}}$) = 0.02 m^3^ mol^−1^, u($V_{\varphi }^{cmc}$) = 0.03 × 10^−3^ m^3^ mol^−1^, and u($\Delta V_{\varphi }^{m}$) = 0.04 × 10^−3^ m^3^ mol^−1^ (level of confidence = 0.68).

**Table 3 pharmaceuticals-16-01187-t003:** Critical micelle concentration (cmc), surface excess concentration ($\Gamma_{\max}$), standard Gibbs free energy of adsorption at interface ($\Delta G^{o}{}_{ad}$), and the minimum surface area per molecule ($A_{\min}$) of SDS in 0.05 mol kg^−1^ DA + ethanol at *T* = (298.15 to 313.15) K and 0.1 M Pa pressure.

Surface Tension Parameters				
cmc/(mol kg^−1^)	0.00772	0.00778	0.00798	0.00805
$\Gamma_{\max}$ 10^6^ (mol m^−2^)	0.44	0.41	0.38	0.35
$A_{\min}$ (nm^2^ mol^−1^)	3.75	4.06	4.32	4.76
$\Pi_{cmc}$ (mN m^−1^)	5.90	5.70	5.69	5.35
$\Delta G^{o}{}_{m}$ (kJ mol^−1^)	−19.69	−20.00	−20.26	−20.57
$\Delta G^{o}{}_{ad}$ (kJ mol^−1^)	−33.00	−33.95	−35.08	−35.89
$\Delta G_{\min}$ (kJ mol^−1^)	36.28	38.58	39.57	42.60

Standard uncertainties, u, are u(*T*) = 0.01 K, u(*p*) = 0.002 MPa, u(^a^m) = 0.0005 mol kg^−1^, u(cmc) = 0.2 × 10^−3^ mol kg^−1^, u($\Gamma_{\max}$) = 0.03 × 10^6^ mol m^−2^, u($A_{\min}$) = 0.04 nm^2^ mol^−1^ and u($\Pi_{cmc}$) = 0.02 mN m^−1^, u($\Delta G^{o}{}_{m}$) = 0.03 kJ mol^−1^, u($\Delta G^{o}{}_{ad}$) = 0.05 kJ mol^−1^ and u($\Delta G_{\min}$) = 0.04 kJ mol^−1^ (level of confidence = 0.68).

**Table 4 pharmaceuticals-16-01187-t004:** Chemical sample description.

Compound	Molecular Formula	Molar Weight (g/mol)	CAS Number	Supplier	Mass Fraction Purity	Purification Method	Purity (after Purification	Water Content (%)
Dopamine hydrochloride	(OH)_2_C_6_H_3_CH_2_CH_2_NH_2_.HCl	189.64	62-31-7	Merck	0.99 ^a^	Used as received	Used as received	
Sodium dodecyl sulfate	C_12_H_25_OSO_2_ONa	288.37	151-21-3	Merck	0.99 ^a^	Used as received	Used as received	
Ethanol	C_2_H_5_OH	46.07	64-17-5	Merck	0.999 ^a^	Distillation	0.999 ^b^	≤0.01 ^c^

^a^ As stated by the supplier. ^b^ GC analysis. ^c^ Measured by Karl Fisher titration (Supplier).

## Data Availability

Data is contained within the article or Appendix A.

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
