# Peer review of "Thermodynamic Insights of the Molecular Interactions of Dopamine (Neurotransmitter) with Anionic Surfactant in Non-Aqueous Media"

_pharmaceuticals, 2023, doi:10.3390/ph16091187_

Round 1

Reviewer 1 Report

The paper submitted by Nabi et al. investigates the micellization of an anionic surfactant (SDS) in the presence of dopamine in ethanol. 

The paper is clear, well written and the conclusions are supported by the results. However, some corrections are needed:

1. the introduction and the discussion sections must be completed with some new references, such as: https://doi.org/10.1002/app.45313; https://doi.org/10.1039/C4SM02766C

2. for figure 2 it would be interesting to show the breakpoints on the figure

3. more specific results may be provided in the conclusion section

The English language is quite poor in some paragraphs. The authors must revise it carefully. 

Author Response

Comment 1: the introduction and the discussion sections must be completed with some new references, such as: https://doi.org/10.1002/app.45313; https://doi.org/10.1039/C4SM02766C

Response:   As per suggestion of reviewer the above papers were cited in Introduction portion.

Comment 2: for figure 2 it would be interesting to show the breakpoints on the figure

Response: Figure 2 is edited; the breakpoints are shown with the crossing lines in revised manuscript.

Comment 3 : more specific results may be provided in the conclusion section

Response:   More specific results are provided in conclusion section as highlighted in red color as per suggestion of reviewer.

Reviewer 2 Report

The present work reported the interactions prevailing in an anionic surfactant, sodium dodecyl sulfate and dopamine hydrochloride in an alcoholic (ethanol) media. The results could shed some lights in revealing the microscopic mechanism of pharmaceuticals. However, there several issues should be addressed before the further consideration of this paper.

1.      The symbols in the abstract,  D G, V, etc., are meaningless.

2.      Most references are several years ago. More recent works should be reviewed. For instance,

[1] J. Pan, et al. Materials 2022, 15(11), 3925

[2] T. Peng, et al. Entropy 2017, 19(11), 620

[3] P. Adhikari, et al. Materials 2021, 14(19), 5774

3.      How many times are repeated for each test? The error bars are missing in some figures.

4.      Why the temperatures 298.15, 303.15, 308.15 and 313.15 K are employed? And what is the pressure during the testing?

fine

Author Response

Comment 1: The symbols in the abstract, D G, V, etc., are meaningless.

Response:  We do not represent individually D, G, V symbols in abstract but delta Go represents standard Gibbs free energy and delta Vo represents the apparent molar volumes.

Comment 2 : Most references are several years ago. More recent works should be reviewed. For instance,

[1] J. Pan, et al. Materials 2022, 15(11), 3925

[2] T. Peng, et al. Entropy 2017, 19(11), 620

[3] P. Adhikari, et al. Materials 2021, 14(19), 5774

Response:   As per suggestion, the most recent related citations are added in the last lines of first paragraph as highlighted in red lines in revised manuscript.

Comment 3: How many times are repeated for each test? The error bars are missing in some figures.

Response:  At least three successive measurements were carried out for each sample until the values are reproducible. We already mentioned error bars in tables and in figures the error bars are not represented.

Comment 4 : Why the temperatures 298.15, 303.15, 308.15 and 313.15 K are employed? And what is the pressure during the testing?

Response:  This is the most acceptable range of drug interaction studies as adopted in many studies concerned with drug-surfactant interactions. Keeping the physiological temperature i.e., 37 oC and drug stability temperature mostly between 25 -40 oC in mind this range of temperature is chosen for thermodynamic micellization process.  The pressure in this study was atmospheric pressure 1 bar or 0.1 M Pa pressure.

Reviewer 3 Report

Comments on pharmaceuticals-2487279:

I consider the current manuscript a round work that fits the scope of the journal. The writing style is good, the experiment and analysis are well-designed and performed, and the deposited data are presented generally in a satisfactory way. The manuscript could be accepted after minor revision, considering the detailed comments listed below.

Table 1 and Figure 1 seem duplicate. For example, the molecular formula in Table 1 and 2D chemical structures in Figure 1 are giving the same information. The same applies to the molecular mass given in Table 1 and molecular weight in Figure 1.  

The authors are using M. Wt in Figure 1 to represent the weight of molecule, while in Table 1 they use molecular mass. For consistency, I would recommend only keep the weight keyword.

The data presented in Table 1 are rather technical and are not really involved in scientifically relevant discussions. Therefore, I would recommend to remove it from the main article.

Some issues of inconsistent formatting exist throughout the paper. For example, the authors are using Fig. and Figure in the same text. For consistency, only a single form should be used. For some references such as ref. 56-57, the citations in text are marked red, which does not seem proper. 

Generally fine. 

Author Response

Comment 1: Table 1 and Figure 1 seem duplicate. For example, the molecular formula in Table 1 and 2D chemical structures in Figure 1 are giving the same information. The same applies to the molecular mass given in Table 1 and molecular weight in Figure 1.

Response: This is very true, to increase visibility and make figure more attractive we show both normal and 2D chemical structure in Figure 1. The molar mass in Table 1 is changed in to Molecular weight as per suggestion of reviewer in revised version.

Comment 2 : The authors are using M. Wt in Figure 1 to represent the weight of molecule, while in Table 1 they use molecular mass. For consistency, I would recommend only keep the weight keyword.

Response: As per suggestion of reviewer, we change molecular mass into molecular weight in Table 1 for consistency as highlighted in red color in revised version.

Comment 3 : The data presented in Table 1 are rather technical and are not really involved in scientifically relevant discussions. Therefore, I would recommend to remove it from the main article.

Response: As Table 1, provides the detailed information of used chemical in present study which is an essential part of information and is mostly seen the various academic research papers, so we wish to present the table 1 in main article.

Comment 4: Some issues of inconsistent formatting exist throughout the paper. For example, the authors are using Fig. and Figure in the same text. For consistency, only a single form should be used. For some references such as ref. 56-57, the citations in text are marked red, which does not seem proper.

Response The reviewer has highlighted the main typological errors, we appreciate the reviewer suggestions and in revised version all of the typological errors are corrected for consistency and the red marked references are updated meanwhile some additional related references are added as highlighted in red color in reference section of revised manuscript.